# Fibrinogen and colorectal cancer: A study of the Chinese population

Jinfeng Wang[1⊗], Jingjing Li[2⊗], Jing Huo[3], Yaxi Zhao[3], Kangkang Liu[2], Huijie Wang[4]*‡, Xu Cao[4]*‡

1 Department of Surgery, Shijiazhuang Traditional Chinese Medicine Hospital, Shijiazhuang, China, 2 Department of Gastroenterology, Shijiazhuang Traditional Chinese Medicine Hospital, Shijiazhuang, China, 3 Department of Endoscopy, Pingshan Traditional Chinese Medicine Hospital, Pingshan County, Shijiazhuang, China, 4 Department of Endoscopy, Shijiazhuang Traditional Chinese Medicine Hospital, Shijiazhuang, China

⊗ These authors contributed equally to this work.
‡ HW and XC also contributed equally to this work.
* Wang_huijie@outlook.com (HW); 18531128959@163.com (XC)

## Abstract

Fibrinogen has been associated with a variety of malignancies, including colorectal cancer (CRC). However, there is no study on the relationship between fibrinogen and CRC specifically in the Chinese population. The objective of this study was to analyze the relationship between fibrinogen and CRC in the Chinese population. This study included consecutive inpatients undergoing colonoscopies (April 2015–June 2022). A total of 3,595 individuals, comprising 468 CRCs and 3,127 controls, were included. Logistic regression models and restricted cubic spline analysis were employed to assess the association between fibrinogen and CRC. When fibrinogen levels were divided into quartiles, the ORs for CRC in Q2, Q3, and Q4 compared to Q1 were 2.72 (95% CI: 1.57–4.74), 4.84 (95% CI: 2.84–8.25), and 9.55 (95% CI: 5.55–16.43), respectively. Restricted cubic spline analysis identified a non-linear association between fibrinogen levels and CRC risk with a threshold of 3.794 (95% CI: 3.747–3.841) g/L. Below this threshold, the CRC risk significantly increased (OR 5.944, 95% CI 4.084–8.65, $P < 0.001$). These findings may offer new insights for the diagnosis and management of CRC.

## Introduction

Colorectal cancer (CRC) ranks as the third most common malignancy worldwide in terms of incidence and second most common in terms of mortality [1]. Despite the availability of effective and resource-sensitive preventive and therapeutic interventions for CRC within the healthcare system [2], the incidence of new CRC cases and related mortality rates have been steadily rising [1].

Recent research has highlighted the frequent association of coagulation disorders with the onset, progression, or treatment of many malignancies, including CRC [3].

**Data availability statement:** Data cannot be shared publicly because of privacy and ethical restrictions. Data are available from the Shijiazhuang Traditional Chinese Medicine Hospital Institutional Data Access/Ethics Committee for researchers who meet the criteria for access to confidential data. For data access requests, please contact the Institutional Review Board at Shijiazhuang Traditional Chinese Medicine Hospital using the following details: Institutional Body: Medical Ethics Committee of Shijiazhuang Traditional Chinese Medicine Hospital Contact Person: Shiyuan Li (Research Service Office) Email Address: kyglzx9016@126.com Phone Number: +86 311 68009016 Institutional Address: Shijiazhuang Traditional Chinese Medicine Hospital, NO. 233 West Zhongshan Road, Shijiazhuang, China.

**Funding:** "The present study was supported by the Hebei Administration of Traditional Chinese Medicine (grant number 2023145). Two members of the research team associated with this grant were involved in conceiving and designing the study, as well as in writing and revising the manuscript."

**Competing interests:** The authors have declared that no competing interests exist.

Virtually all types of cancer are linked to a hypercoagulable state [4], which manifests as coagulation activation in cancer patients—a subclinical condition referred to as cancer-associated hypercoagulability [5]. The interplay between cancer and hemostasis is bidirectional [6]: cancer cells can activate components of the hemostatic system through multiple mechanisms [7], thereby creating a hypercoagulable microenvironment through the expression of procoagulants, tissue factors, or inflammatory cytokines [8]. Conversely, elements of the hemostatic system can actively contribute to cancer pathogenesis [7], promoting malignant proliferation, invasion, and metastasis [3]. In investigations into the correlation between CRC and coagulation indicators, platelet count, D-dimer, and fibrinogen have been identified as factors associated with CRC progression and prognosis [9–16].

Fibrinogen, an acute-phase protein, exhibits varying degrees of abnormalities in individuals experiencing surgery, infection, trauma, tumors, and coagulation-related disorders [17–22]. Extensive research has focused on the role of this glycoprotein. Fibrinogen, a key inflammatory regulator, has been implicated in processes such as the formation of an inflammatory microenvironment, angiogenesis, malignant cell proliferation, invasion, and metastasis [23]. Studies have demonstrated that CRC growth and metastasis are dependent on the involvement of fibrinogen [24], and elevated preoperative fibrinogen levels have been linked to poor CRC prognosis [25–29]. Nevertheless, limited research has explored the relationship between fibrinogen and CRC in the general populace [14,16], especially among Chinese individuals.

Exploring the risk factors for CRC can contribute to the development of more detailed, noninvasive colorectal screening methods [30]. The objective of this research was to analyze the relationship between fibrinogen levels and CRC in the Chinese population.

## Materials and methods

This retrospective study was approved by the Medical Ethics Committee of Shijiazhuang Traditional Chinese Medicine Hospital (NO.20220919029), and the requirements for informed consent were waived due to the retrospective nature.

The study involved the enrollment of 3595 consecutive inpatients from April 2015 to June 2022. All individuals underwent colonoscopy, and each patient was included only once to ensure exclusivity. Histopathological reports confirmed the presence of colorectal cancer (initial diagnosis). Subjects with normal colonoscopy results were considered as controls (for details, see Fig 1). The data for this study were accessed on September 22, 2022.

Clinical (including comorbidities and laboratory data) and demographic information were derived from clinical records. Laboratory data were selected from the first qualifying report prior to colonoscopy during hospitalization. The definition of liver disease was based on a previous study [31].

The statistical analyses were performed using R 3.3.2 (http://www.R-project.org, The R Foundation) and Free Statistics software version 1.7.1 (http://www.clinicalscientists.cn/freestatistics, Beijing, China). Significance was determined at a $P$ value$<0.05$ (2-sided).

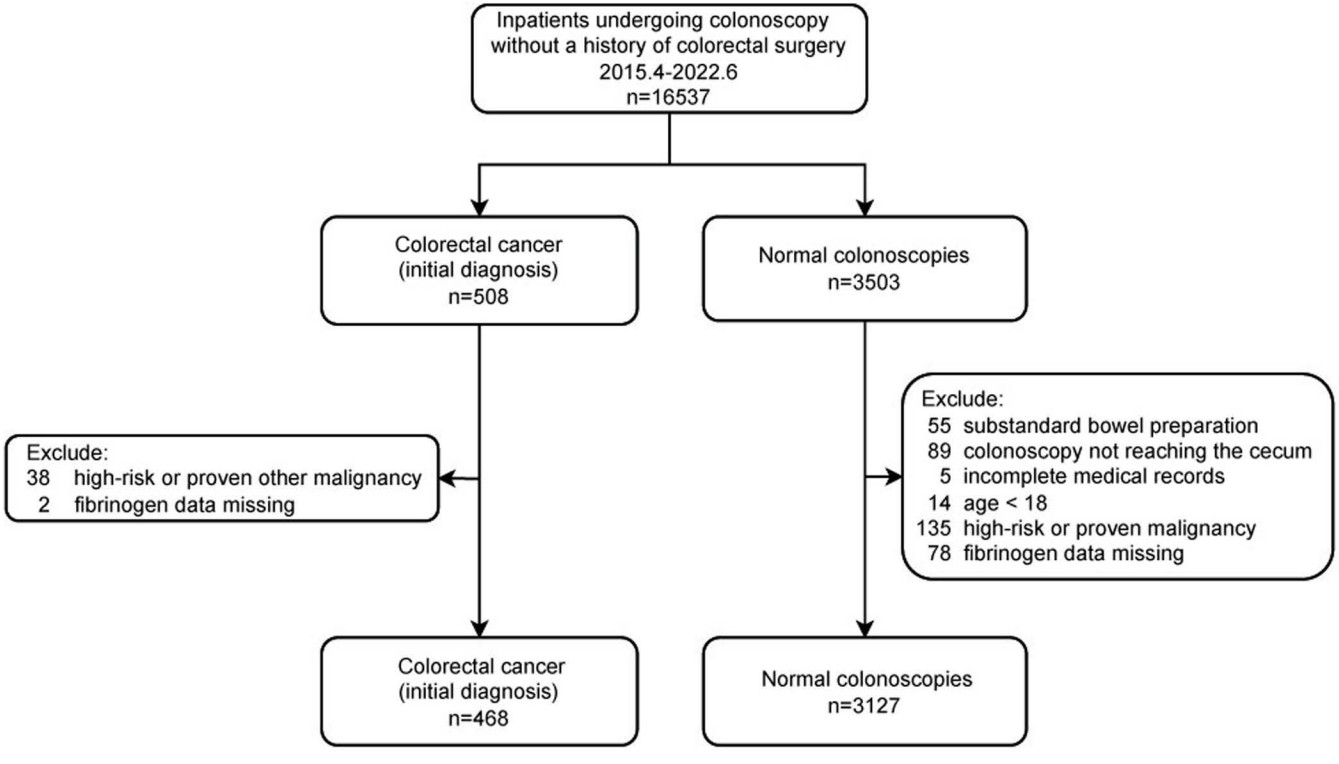

**Fig 1. The flow chart.**

The study data utilized Mean ± standard deviation or median (Q1–Q3) values based on the normality of the distribution. Statistical tests such as the Mann–Whitney $U$ test or Student's $t$ test were used for continuous variables. In contrast, Fisher's exact test or the chi-squared test were used for categorical variables.

To analyze the relationship between fibrinogen and CRC, we used multivariate binary logistic regression and restricted cubic spline analysis. Three models were constructed in multivariate binary logistic regression: (a) unadjusted; (b) adjusted for age and sex; (c) additionally adjusted for weight, family history of CRC, drinking and smoking status, total bilirubin (TBIL), aspartate aminotransferase (AST), alanine aminotransferase (ALT), albumin (ALB), gamma-glutamyl transferase (GGT), cholinesterase (ChE), alkaline phosphatase (ALP), total bile acid (TBA), glucose (GLU), creatinine (CREA), urea, low-density lipoprotein (LDL), high-density lipoprotein (HDL), total protein (TP), lipoprotein(a) [Lp(a)], platelets (PLT), white blood cell (WBC), neutrophil count, activated partial thromboplastin time (APTT), PT-INR, thrombin time (TT), hypertension, and liver disease. These potential confounders were chosen based on a change of ≥ 10% in effect estimates or significant covariates in Table 1 ($P < 0.05$). Subgroup analysis was conducted using a stratified binary logistic regression followed by a test for interaction. Missing data were handled using multiple imputations by chained equations with five replications. In addition, a sensitivity analysis using complete cases was performed to assess the robustness of the results.

## Results

### Participants

Table 1 shows the baseline features of 3595 individuals (468 CRCs and 3127 controls). Various characteristics included age, sex, weight, smoking status, drinking status, fibrinogen, TBIL, TP, AST, ALT, ALB, GGT, ChE, ALP, TBA, GLU, CREA,

**Table 1. Participants characteristics.**

| Variables | Total n = 3595 | Case n = 468 | Control n = 3127 | P-value |
|---|---|---|---|---|
| Age, yr | 52.4 ± 13.4 | 64.8 ± 11.1 | 50.5 ± 12.8 | < 0.001 |
| Sex, male | 1452 (40.4) | 266 (56.8) | 1186 (37.9) | < 0.001 |
| Weight, kg | 66.8 ± 12.3 | 67.9 ± 11.2 | 66.6 ± 12.4 | 0.031 |
| Marital status | | | | 0.746 |
| Single/ divorced | 168 (4.7) | 24 (5.1) | 144 (4.6) | |
| Married | 3237 (90.0) | 422 (90.2) | 2815 (90.0) | |
| Others | 190 (5.3) | 22 (4.7) | 168 (5.4) | |
| Smoking status | | | | 0.191 |
| Non-smoker | 3267 (64.2) | 421(90) | 2846 (91) | |
| Current smoker | 257 (7.1) | 41 (8.8) | 216 (6.9) | |
| Ex-smoker | 71 (2.0) | 6 (1.3) | 65 (2.1) | |
| Drinking status | | | | 0.973 |
| Non-drinker | 3252 (90.5) | 424 (90.6) | 2828 (90.4) | |
| Current drinker | 292 (8.1) | 37 (7.9) | 255 (8.2) | |
| Ex-drinker | 51 (1.4) | 7 (1.5) | 44 (1.4) | |
| Family history of CRC | 46 (1.3) | 4 (0.9) | 42 (1.3) | 0.381 |
| Fibrinogen, g/L | 2.9 ± 0.7 | 3.5 ± 0.8 | 2.8 ± 0.7 | < 0.001 |
| ALT, U/L | 17.0 (12.0, 25.0) | 14.0 (10.7, 19.0) | 17.0 (13.0, 25.9) | < 0.001 |
| AST, U/L | 19.0 (16.0, 23.5) | 18.0 (15.0, 21.0) | 19.8 (16.9, 24.0) | < 0.001 |
| ALB, g/L | 44.1 ± 3.8 | 41.8 ± 4.3 | 44.5 ± 3.6 | < 0.001 |
| GGT, U/L | 18.0 (14.0, 27.0) | 19.0 (14.0, 28.0) | 18.0 (14.0, 27.0) | 0.038 |
| ChE, U/L | 8730.8 ± 1987.6 | 7591.8 ± 1870.4 | 8901.3 ± 1948.4 | < 0.001 |
| ALP, U/L | 75.6 ± 28.3 | 87.7 ± 45.5 | 73.8 ± 24.2 | < 0.001 |
| TBIL, µmol/L | 12.6 (9.7, 16.4) | 11.7 (8.5, 15.0) | 12.7 (9.8, 16.6) | < 0.001 |
| DBIL, µmol/L | 2.7 (2.0, 3.8) | 2.7 (2.0, 3.8) | 2.7 (2.0, 3.7) | 0.349 |
| TBA, µmol/L | 2.6 (1.4, 4.4) | 3.0 (1.5, 5.0) | 2.5 (1.4, 4.3) | 0.003 |
| GLU, mmol/L | 6.0 ± 1.9 | 6.8 ± 2.6 | 5.9 ± 1.7 | < 0.001 |
| Urea, mmol/L | 4.7 ± 1.5 | 5.1 ± 1.9 | 4.7 ± 1.4 | < 0.001 |
| CREA, µmol/L | 63.5 ± 19.8 | 69.5 ± 33.4 | 62.7 ± 16.6 | < 0.001 |
| UA, µmol/L | 300.3 ± 93.4 | 301.8 ± 86.4 | 300.1 ± 94.4 | 0.707 |
| TP, g/L | 71.8 ± 5.4 | 69.6 ± 6.1 | 72.1 ± 5.2 | < 0.001 |
| TC, mmol/L | 4.9 ± 1.0 | 4.8 ± 1.0 | 4.9 ± 1.0 | 0.002 |
| TG, mmol/L | 1.3 (0.9, 1.9) | 1.3 (0.9, 1.7) | 1.3 (0.9, 1.9) | 0.705 |
| HDL, mmol/L | 1.4 ± 0.3 | 1.3 ± 0.3 | 1.4 ± 0.3 | < 0.001 |
| LDL, mmol/L | 2.9 ± 0.6 | 2.8 ± 0.6 | 2.9 ± 0.6 | 0.051 |
| Apo B, g/L | 1.0 ± 0.2 | 1.0 ± 0.2 | 1.0 ± 0.2 | 0.294 |
| Lp(a), mg/L | 141.0 (78.0, 293.5) | 200.0 (100.0, 381.8) | 133.0 (75.2, 280.4) | < 0.001 |
| PLT, × 10⁹/L | 241.5 ± 64.1 | 265.2 ± 87.5 | 237.9 ± 59.1 | < 0.001 |
| WBC, × 10⁹/L | 5.9 ± 1.9 | 6.7 ± 2.2 | 5.7 ± 1.8 | < 0.001 |
| Neutrophil count, × 10⁹/L | 3.8 ± 1.7 | 4.6 ± 2.0 | 3.6 ± 1.6 | < 0.001 |
| TT, (s) | 16.3 ± 1.9 | 15.9 ± 2.4 | 16.4 ± 1.8 | < 0.001 |
| PT-INR | 1.0 ± 0.1 | 1.0 ± 0.1 | 1.0 ± 0.1 | < 0.001 |
| PT, s | 12.1 ± 1.2 | 12.3 ± 1.4 | 12.0 ± 1.2 | < 0.001 |
| APTT, s | 29.5 ± 5.6 | 30.0 ± 6.1 | 29.4 ± 5.5 | 0.050 |
| Co-morbidities | | | | |

*(Continued)*

**Table 1.** (Continued)

| Variables | Total<br>n = 3595 | Case<br>n = 468 | Control<br>n = 3127 | P-value |
|---|---|---|---|---|
| Hypertension | 852 (23.7) | 177 (37.8) | 675 (21.6) | < 0.001 |
| Ischemic cerebrovascular disease | 349 (9.7) | 56 (12.0) | 293 (9.4) | 0.077 |
| DM | 620 (17.2) | 147 (31.4) | 473 (15.1) | < 0.001 |
| CHD | 383 (10.7) | 61 (13.0) | 322 (10.3) | 0.073 |
| HLP | 310 (8.6) | 18 (3.8) | 292 (9.3) | < 0.001 |
| Liver disease | 375 (10.4) | 30 (6.4) | 345 (11.0) | 0.002 |

Data presented are mean ± SD, median (quartile 1–quartile 3), or N (%).

Abbreviations: CRC, colorectal cancer; ALT, alanine aminotransferase; AST, aspartate aminotransferase; ALP, alkaline phosphatase; ALB, albumin; GGT, gamma-glutamyl transferase; ChE, cholinesterase; TBIL, total bilirubin; DBIL, direct bilirubin; TBA, total bile acid; GLU, glucose; CREA, creatinine; UA, uric acid; TP, total protein; TC, total cholesterol; TG, triglyceride; LDL, low-density lipoprotein; HDL, high-density lipoprotein; Apo B, apolipoprotein B; Lp(a), lipoprotein(a); PLT, platelets; WBC, white blood cell; TT, thrombin time; PT-INR, the international normalized ratio of prothrombin time; PT, pro-thrombin time; APTT, activated partial thromboplastin time; DM, diabetes mellitus; HLP, hyperlipemia; CHD, coronary heart disease.

urea, TC, HDL, Lp(a), PLT, WBC, neutrophil count, APTT, TT, PT-INR, PT, hypertension, HLP, liver disease, and DM (*P*<0.05).

## Fibrinogen and colorectal cancer

Table 2 presents a significant relationship between fibrinogen and CRC in the multivariate logistic regression analyses. In the crude model, fibrinogen was strongly associated with CRC (odds ratio [OR] = 3.04 [2.66–3.46]); adjusted for age and sex, the OR was 2.48 (2.15–2.86); and adjusted for the full model, the OR was 1.83 (1.52–2.20). After transforming fibrinogen into a categorical variable, fibrinogen was positively associated with CRC in different models, adjusting for other potential confounding factors ( *P*<0.001).

The restricted cubic spline analysis (Fig 2, 99.5% data) revealed a non-linear relationship between fibrinogen levels and CRC risk, identifying a threshold at 3.794 (95% CI: 3.747–3.841) g/L. Specifically, below this threshold, the CRC risk

**Table 2. Multivariable logistic regression analyses of fibrinogen and CRC.**

| Variable | Event, n (%) | Unadjusted | | Model I[a] | | Model II[b] | |
|---|---|---|---|---|---|---|---|
| | | OR (95% CI) | P-value | OR (95% CI) | P-value | OR (95% CI) | P-value |
| Fibrinogen, g/L | 468/3595(13.0) | 3.04 (2.66~3.46) | <0.001 | 2.48 (2.15~2.86) | <0.001 | 1.83 (1.52~2.20) | <0.001 |
| Fibrinogen quartiles, g/L | | | | | | | |
| Q1 (<2.43) | 21/896 (2.3) | 1(ref.) | | 1(ref.) | | 1(ref.) | |
| Q2 (2.43–2.79) | 57/893 (6.4) | 2.84 (1.71~4.73) | <0.001 | 2.7 (1.59~4.6) | <0.001 | 2.72 (1.57~4.74) | <0.001 |
| Q3 (2.80–3.24) | 108/905(11.9) | 5.65 (3.50~9.10) | <0.001 | 4.78 (2.89~7.89) | <0.001 | 4.84 (2.84~8.25) | <0.001 |
| Q4 (≥3.25) | 282/901(31.3) | 18.98 (12.04~29.92) | <0.001 | 13.4 (8.3~21.65) | <0.001 | 9.55 (5.55~16.43) | <0.001 |
| *P* for trend | | | <0.001 | | <0.001 | | <0.001 |

OR, odds ratio; CI, confidence interval; Q, quartile; ALB, albumin; GGT, gamma-glutamyl transferase; ALP, alkaline phosphatase; AST, aspartate amino-transferase; ALT, alanine aminotransferase; ChE, cholinesterase; TBA, total bile acid; TBIL, total bilirubin; HDL, high-density lipoprotein; LDL, low-density lipoprotein; GLU, glucose; CREA, creatinine; TP, total protein; Lp(a), lipoprotein(a); PLT, platelets; WBC, white blood cell; PT-INR, international normal-ized ratio of prothrombin time; TT, thrombin time; APTT, activated partial thromboplastin time; CRC, colorectal cancer.

[a]Adjusted for age and sex. [b]Adjusted for Model I+weight, family history of CRC, drinking status, smoking status, ALB, ALT, AST, ALP, GGT, ChE, TBIL, TBA, GLU, CREA, urea, HDL, LDL, TP, Lp(a), PLT, WBC, neutrophil count, PT-INR, APTT, TT, hypertension, and liver disease.

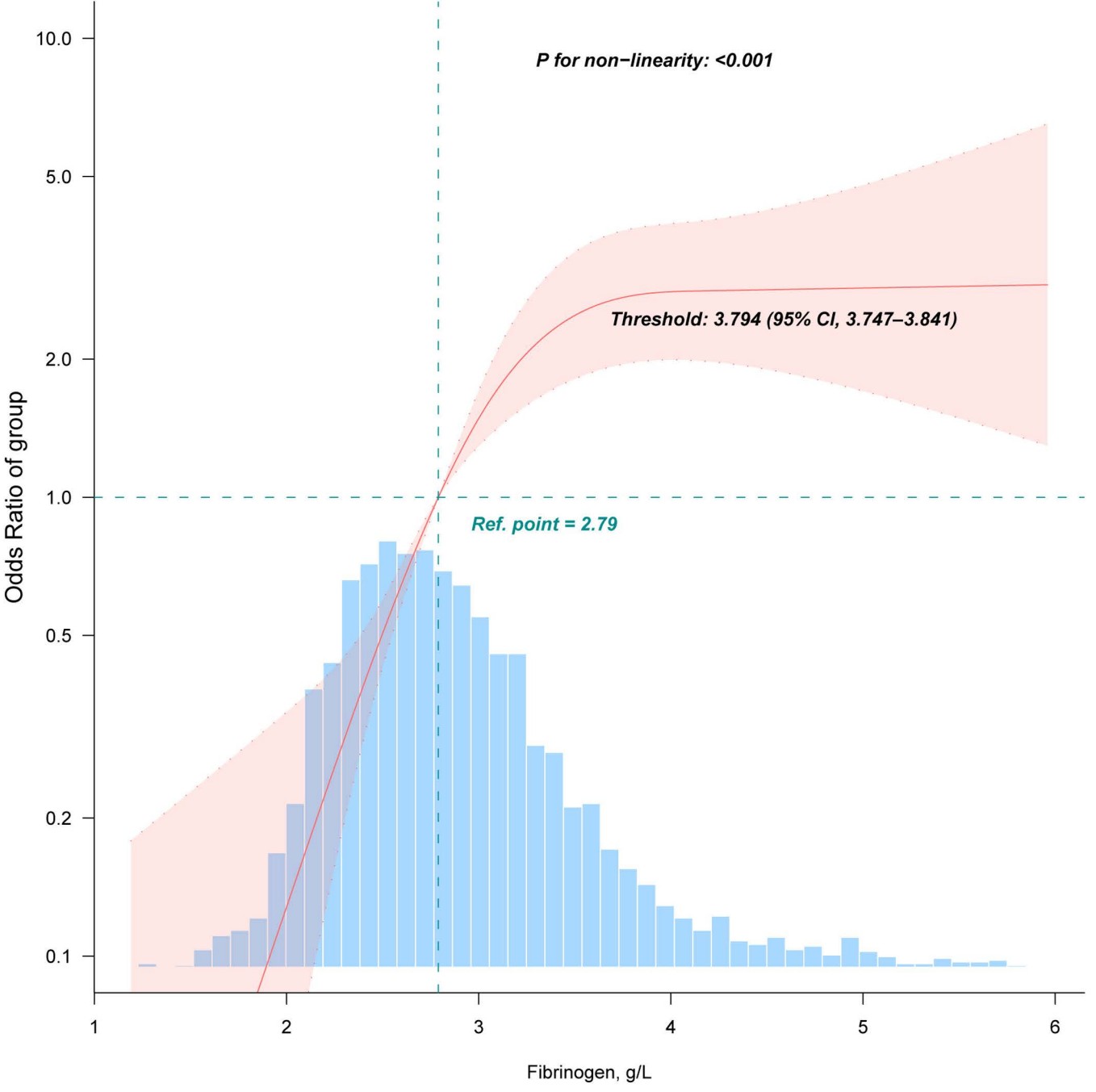

**Fig 2. Association between fibrinogen and colorectal cancer.** Adjusted for age, sex, weight, family history of colorectal cancer, drinking status, smoking status, albumin, gamma-glutamyl transferase, alanine aminotransferase, alkaline phosphatase, aspartate aminotransferase, cholinesterase, total bilirubin, total bile acid, glucose, creatinine, urea, total protein, low-density lipoprotein, high-density lipoprotein, lipoprotein(a), platelets, white blood cell, neutrophil count, the international normalized ratio of prothrombin time, activated partial thromboplastin time, thrombin time, hypertension, and liver disease. Only 99.5% of the data is shown.

**Table 3. Threshold effect analysis of the relationship of fibrinogen and CRC.**

| Fibrinogen, g/L | Adjusted Model | |
|---|---|---|
| | OR (95% CI) | *P*-value |
| <3.794 | 5.944 (4.084~8.65) | <0.001 |
| ≥3.794 | 0.838 (0.431~1.632) | 0.604 |
| Likelihood Ratio test | | <0.001 |

OR, odds ratio; CI, confidence interval; ALB, albumin; GGT, gamma-glutamyl transferase; ALP, alkaline phosphatase; AST, aspartate aminotransferase; ALT, alanine aminotransferase; ChE, cholinesterase; TBA, total bile acid; TBIL, total bilirubin; HDL, high-density lipoprotein; LDL, low-density lipoprotein; GLU, glucose; CREA, creatinine; TP, total protein; Lp(a), lipoprotein(a); PLT, platelets; WBC, white blood cell; PT-INR, international normalized ratio of prothrombin time; TT, thrombin time; APTT, activated partial thromboplastin time; CRC, colorectal cancer.

Adjusted for age and sex, weight, family history of CRC, drinking status, smoking status, ALB, ALT, AST, ALP, GGT, ChE, TBIL, TBA, GLU, CREA, urea, HDL, LDL, TP, Lp(a), PLT, WBC, neutrophil count, PT-INR, APTT, TT, hypertension, and liver disease.

significantly increases (OR 5.944, 95% CI 4.084–8.65, *P*<0.001), whereas, above this threshold, the association is no longer statistically significant (Table 3, OR = 0.838, 95% CI: 0.431–1.632, *P* = 0.604).

## Sensitivity analysis

To explore the stability of the association between fibrinogen and CRC, we performed subgroup analyses. The relationship between fibrinogen and CRC was not significantly affected by subgroups such as sex, age, HLP, hypertension, ischemic cerebrovascular disease, and CHD (*P* for interaction >0.05, Fig 3).

In the sensitivity analysis, participants with missing data were not included, leaving 1695 individuals. S1 Table showed that the relationship between fibrinogen and CRC was stable. In the crude model, fibrinogen was associated with CRC (OR = 3.06, 95% CI: 2.55–3.68); in Model I (adjustment for age and sex), the OR was 2.54 (2.08–3.10); and in Model II (adjustment for full model), the OR was 1.67 (1.27–2.21). The significant relationship also existed in different adjustment models when fibrinogen was transformed into quartiles (*P* for trend <0.001).

## Discussion

In this case-control study, we found that fibrinogen levels are positively correlated with CRC risk. Subgroup analyses further confirmed the stability of this association. Notably, we identified an inverse L-shaped relationship between fibrinogen levels and CRC risk, with a significant threshold at 3.794 g/L.

Our findings are in line with previous investigations exploring the relationship between fibrinogen and CRC in the general population. These studies have consistently suggested that elevated levels of fibrinogen are independently and positively associated with CRC risk [14,16]. For example, a prospective study involving 84,000 individuals from the general population in Denmark found that high levels of fibrinogen were associated with an increased risk of CRC. Similarly, research from the Moli-Sani cohort indicated that individuals. These studies suggest that fibrinogen may be an important biomarker for assessing CRC risk, potentially offering new approaches and methods for the early diagnosis and treatment of CRC.

Beyond the hemostatic factors, we meticulously explored the potential confounders, including biomarkers associated with CRC or other cancers identified in prior studies (e.g., UA, AST, ALT, GLU, HDL, and PLT) [32–36]. Additionally, we considered other potential confounders such as biochemical indicators, comorbidities, and family history. Remarkably, these factors had only minimal impact on the relationship between fibrinogen and CRC, underscoring the robustness of our findings.

Although this retrospective observational study cannot confirm a causal relationship between fibrinogen and CRC, evidence from both basic and clinical research indicates an interaction between cancer and the hemostatic system. Cancer may promote thromboembolism, while pathological activation of the hemostatic system can drive cancer progression. The thrombin-fibrinogen axis plays a crucial role in CRC [37]. Fibrinogen interacts with leukocyte integrin αMβ2,

| Subgroup | Total | Event (%) | OR (95%CI) | | P for interaction |
|---|---|---|---|---|---|
| **Sex** | | | | | |
| Female | 2143 | 202 (9.4) | 1.58 (1.19~2.11) | | 0.124 |
| Male | 1452 | 266 (18.3) | 2.05 (1.59~2.65) | | |
| **Age, yr** | | | | | |
| <65 | 2889 | 221 (7.6) | 1.91 (1.5~2.44) | | 0.296 |
| >=65 | 706 | 247 (35) | 1.8 (1.32~2.46) | | |
| **Hypertension** | | | | | |
| No | 2743 | 291 (10.6) | 1.89 (1.5~2.39) | | 0.283 |
| Yes | 852 | 177 (20.8) | 1.89 (1.35~2.64) | | |
| **Ischemic cerebrovascular disease** | | | | | |
| No | 3246 | 412 (12.7) | 1.85 (1.51~2.26) | | 0.255 |
| Yes | 349 | 56 (16) | 2.2 (1.05~4.63) | | |
| **CHD** | | | | | |
| No | 3212 | 407 (12.7) | 1.7 (1.4~2.08) | | 0.055 |
| Yes | 383 | 61 (15.9) | 3.93 (1.73~8.92) | | |
| **HLP** | | | | | |
| No | 3285 | 450 (13.7) | 1.8 (1.49~2.18) | | 0.339 |
| Yes | 310 | 18 (5.8) | 4.72 (0.9~24.8) | | |

Effect(95%CI): 1.0 2.0 4.0 8.0 16.0

**Fig 3. Association between fibrinogen and colorectal cancer.** Except for the subgroup component itself, each subgroup factor was adjusted for age, sex, weight, family history of colorectal cancer, drinking status, smoking status, albumin, gamma-glutamyl transferase, alanine aminotransferase, alkaline phosphatase, aspartate aminotransferase, cholinesterase, total bilirubin, total bile acid, glucose, creatinine, urea, total protein, low-density lipoprotein, high-density lipoprotein, lipoprotein(a), platelets, white blood cell, neutrophil count, the international normalized ratio of prothrombin time, activated partial thromboplastin time, thrombin time, hypertension, and liver disease.

promoting inflammatory responses that lead to increased stress and phosphorylation of proliferation markers in intestinal epithelial cells, indicating a direct link between fibrinogen/inflammatory function and tumorigenic changes in colon epithelial cells [38]. Fibrinogen promotes angiogenesis by stimulating endothelial cell migration, proliferation, and tubular structure formation, and it binds growth factors such as bFGF and VEGF, protecting them from degradation and enhancing endothelial cell activation, thereby reinforcing angiogenesis [39]. Additionally, the fibrin matrix plays a key role in platelet stabilization and activation, and platelets promote the expansion and function of immunosuppressive myeloid cells during the early stages of inflammation-induced adenoma formation, and their infiltration into colon adenomas, further driving tumor cell metastasis [40]. These mechanisms collectively promote tumor growth, angiogenesis, and metastasis in CRC patients through the interplay of the inflammatory and coagulation axes. Conversely, tumors can induce hypoxia and

hypercoagulable states in vivo [41,42], leading to elevated fibrinogen levels in CRC patients, which are a result of tissue damage and subsequent inflammatory responses [43].

We discovered a non-linear, inverse L-shaped relationship between fibrinogen levels and CRC risk, identifying a critical threshold at 3.794 g/L. This finding provides strong evidence supporting the use of fibrinogen levels in clinical interventions to reduce CRC risk. Notably, when fibrinogen levels are controlled below 3.794 g/L, CRC risk decreases significantly as fibrinogen levels decline, making this threshold crucial for managing fibrinogen levels. Additionally, the diagnostic utility of fibrinogen may be enhanced when combined with other tumor markers, such as carcinoembryonic antigen and prealbumin, potentially improving diagnostic performance in CRC [44]. Future clinical trials and studies are needed to further explore the role of fibrinogen in the adenoma-carcinoma sequence and validate its effectiveness as a diagnostic biomarker. Upon such validation, monitoring fibrinogen levels could become a valuable risk assessment tool prior to colonoscopy in CRC screening, owing to its efficiency, speed, and non-invasive nature. When elevated fibrinogen levels are detected, they should be assessed in conjunction with other risk factors to evaluate CRC risk, and colonoscopy should be employed for screening high-risk individuals.

Several limitations should be acknowledged in our study. Firstly, missing data is a common issue in retrospective studies. However, we addressed this by employing multiple imputations, and our results remained consistent. Secondly, we lacked long-term data on fibrinogen levels, which could offer valuable insights into its impact on the progression of CRC. Thirdly, despite controlling for various factors, unmeasured lifestyle factors (e.g., diet, physical activity), infections, and socioeconomic variables could still influence fibrinogen levels and CRC risk. Furthermore, our study design, being retrospective, primarily reveals correlation rather than causation. Finally, our study primarily involved Chinese participants, limiting the generalizability of our results to other demographic groups. For greater reliability and wider applicability, it is crucial that future research validates these results with independent cohorts.

## Conclusion

Fibrinogen levels are positively correlated with CRC risk, and the relationship between them exhibits an inverse L-shape, with a critical turning point at 3.794 g/L. Future clinical trials and studies are necessary to further validate the role of fibrinogen in CRC clinical management.

## Supporting information

**S1 Table. Multivariable logistic regression analyses of fibrinogen and colorectal cancer before multiple imputation.**
(DOCX)

## Acknowledgments

We thank Free Statistics team for providing technical assistance and valuable tools for data analysis and visualization.

## Author contributions

**Conceptualization:** Huijie Wang, Xu Cao.

**Data curation:** Jinfeng Wang, Jingjing Li, Jing Huo, Yaxi Zhao, Kangkang Liu.

**Formal analysis:** Huijie Wang, Jinfeng Wang, Jingjing Li, Jing Huo, Yaxi Zhao, Kangkang Liu.

**Investigation:** Jinfeng Wang, Jingjing Li.

**Methodology:** Huijie Wang.

**Writing – original draft:** Huijie Wang, Xu Cao.

**Writing – review & editing:** Huijie Wang, Xu Cao.

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
