## [Decision Letter · Decision Letter 0]

22 Dec 2024

PONE-D-24-36493Fibrinogen and colorectal cancer: A study of Chinese populationPLOS ONE

Dear Dr. Wang,

Thank you for submitting your manuscript to PLOS ONE. After careful consideration, we feel that it has merit but does not fully meet PLOS ONE’s publication criteria as it currently stands. Therefore, we invite you to submit a revised version of the manuscript that addresses the points raised during the review process.

We look forward to receiving your revised manuscript.

Kind regards,

José M. Alvarez-Suarez

Academic Editor

PLOS ONE

“The present study was supported by the Hebei Administration of Traditional Chinese Medicine (grant number 2023145).”

Reviewers' comments:

Reviewer's Responses to Questions

**Comments to the Author**

1. Is the manuscript technically sound, and do the data support the conclusions?

Reviewer #1: Partly

Reviewer #2: Partly

2. Has the statistical analysis been performed appropriately and rigorously? 

Reviewer #1: No

Reviewer #2: Yes

3. Have the authors made all data underlying the findings in their manuscript fully available?

Reviewer #1: No

Reviewer #2: Yes

4. Is the manuscript presented in an intelligible fashion and written in standard English?

Reviewer #1: Yes

Reviewer #2: Yes

5. Review Comments to the Author

Reviewer #1: This manuscript investigates the relationship between fibrinogen levels and colorectal cancer (CRC) risk specifically within the Chinese population. Analyzing data from 3,595 individuals, including 468 CRC cases and 3,127 controls, collected between April 2015 and June 2022, the researchers utilized logistic regression models and smooth curve fitting for assessment. The findings indicate a significant linear association between elevated fibrinogen levels and an increased risk of CRC, with an adjusted odds ratio of 1.83 [1.52–2.20], P < 0.001. Subgroup and sensitivity analyses further support these conclusions, suggesting that this relationship is stable across various demographic and clinical characteristics. The manuscript is clear and logical, but there are some issues that need improvement. I only recommend the manuscript for publication if the authors can carefully revise the following detailed comments in the manuscript:

1. The rationale behind choosing specific statistical models (e.g., logistic regression) over others should be explained more clearly.

2. Have the findings been externally validated using an independent cohort or dataset?

3. Despite adjusting for multiple confounders, the potential impact of unmeasured factors like dietary habits and physical activity cannot be ignored. These variables could significantly influence both fibrinogen levels and CRC risk. The authors should acknowledge this limitation and discuss its implications for their findings.

4. The choice of statistical models (e.g., logistic regression) over others is not adequately justified. The rationale behind selecting specific methods and cut-off points for quartiles must be better explained, along with their clinical relevance.

5. What specific clinical applications or changes in practice could arise from these findings? How do the authors envision their results impacting patient care or public health strategies?

Reviewer #2: This manuscript provides valuable insights into the association between fibrinogen and CRC in the Chinese population. However, several aspects require further elaboration to enhance the robustness and interpretability of the findings.

1. The manuscript sometimes implies a causal relationship between fibrinogen and CRC (e.g., "fibrinogen promotes cancer progression"). However, as this is a cross-sectional and retrospective study, causality cannot be inferred. Reframe the language throughout the manuscript to emphasize association rather than causality.

2. Although the study adjusts for numerous confounders (e.g., BMI, smoking, liver disease), certain lifestyle factors (diet, exercise), acute or chronic infective conditions, and socioeconomic variables are missing. These may influence both fibrinogen levels and CRC risk.

3. The study highlights a linear association between fibrinogen and CRC risk. However, non-linear relationships or threshold effects are not thoroughly investigated. Consider performing and reporting restricted cubic spline analyses to assess potential non-linear associations. This could reveal critical fibrinogen thresholds that significantly elevate CRC risk. Furthermore, if non-linear patterns are observed, discuss how clinical guidelines might use fibrinogen cut-off values to stratify CRC risk in patients.

4. Expand the discussion by integrating more references on the thrombin-fibrinogen axis or tumor microenvironment, and how fibrinogen promotes tumor growth, angiogenesis, and metastasis in CRC patients.

5. Abbreviations like ALP, AST, WBC, etc. used earlier in the text should be consistently applied in the subsequent sensitivity analysis section.

6. The link between fibrinogen and colorectal cancer might be attributed to chronic inflammation and the changes in the body's internal environment caused by tumors, such as coagulation issues. This is similar to other inflammatory markers like C-reactive protein, which also lack specificity. The studies mentioned by the author in the discussion section only show a connection between fibrinogen and colorectal cancer, but they don't establish that high levels of fibrinogen are a risk factor for the disease. Therefore, the author should provide a clearer explanation of the significance of their research. It's widely understood that any malignant tumor can lead to chronic inflammation and is associated with related inflammatory markers.

6. PLOS authors have the option to publish the peer review history of their article (what does this mean? ). If published, this will include your full peer review and any attached files.

**Do you want your identity to be public for this peer review?** For information about this choice, including consent withdrawal, please see our Privacy Policy .

Reviewer #1: No

Reviewer #2: **Yes: ** WEI WANG

---

## [Author Response · Author response to Decision Letter 0]

28 Jan 2025

Dear Editor and reviewers,

On behalf of my co-authors, we greatly appreciate the careful review and comments from both you and the reviewers. We believe that by implementing the suggested changes, we now have a better manuscript entitled “Fibrinogen and colorectal cancer: A study of Chinese population” (ID: PONE-D-24-36493) for submission to PLOS ONE. We look forward to your positive response to the revised work submitted here.

We present here point-to-point responses for each of the comments in the attached document and have revised our manuscript accordingly. We do not change our raw data or results. Some grammatical and writing errors in the manuscript have been corrected. And we hope the revised manuscript could be acceptable to you. Revised sections are identified with red text in the paper.

There are no conflicts of interest regarding this work. All authors have read the revised manuscript and approved its submission to the PLOS ONE. Please do not hesitate to contact us if we can be of any further assistance.

Thank you and best regards.

Yours Sincerely,

Huijie Wang

E-mail: wang_huijie@outlook.com

Shijiazhuang Traditional Chinese Medicine Hospital, NO. 233 Zhongshan West Road, Shijiazhuang, China

Responds to the reviewer’s comments:

Reply to Reviewer #1

Thank you very much for your time involved in reviewing the manuscript and your very encouraging comments on the merits.

Comments:

“This manuscript investigates the relationship between fibrinogen levels and colorectal cancer (CRC) risk specifically within the Chinese population. Analyzing data from 3,595 individuals, including 468 CRC cases and 3,127 controls, collected between April 2015 and June 2022, the researchers utilized logistic regression models and smooth curve fitting for assessment. The findings indicate a significant linear association between elevated fibrinogen levels and an increased risk of CRC, with an adjusted odds ratio of 1.83 [1.52–2.20], P < 0.001. Subgroup and sensitivity analyses further support these conclusions, suggesting that this relationship is stable across various demographic and clinical characteristics. The manuscript is clear and logical, but there are some issues that need improvement. I only recommend the manuscript for publication if the authors can carefully revise the following detailed comments in the manuscript:

1. The rationale behind choosing specific statistical models (e.g., logistic regression) over others should be explained more clearly.

2. Have the findings been externally validated using an independent cohort or dataset?

3. Despite adjusting for multiple confounders, the potential impact of unmeasured factors like dietary habits and physical activity cannot be ignored. These variables could significantly influence both fibrinogen levels and CRC risk. The authors should acknowledge this limitation and discuss its implications for their findings.

4. The choice of statistical models (e.g., logistic regression) over others is not adequately justified. The rationale behind selecting specific methods and cut-off points for quartiles must be better explained, along with their clinical relevance.

5. What specific clinical applications or changes in practice could arise from these findings? How do the authors envision their results impacting patient care or public health strategies?”

Comment 1:

“The rationale behind choosing specific statistical models (e.g., logistic regression) over others should be explained more clearly.”

Response:

Thank you for your valuable comments on our manuscript. We would like to provide the following explanations for choosing the logistic regression model:

1. Alignment with Research Objectives:

Case-control studies aim to explore the association between exposure and disease occurrence. Logistic regression can directly estimate the probability relationship between exposure and disease occurrence, obtaining odds ratios (ORs) that clearly demonstrate the impact of exposure factors on disease For instance, in studies examining the relationship between smoking and lung cancer, logistic regression can intuitively determine how much smoking increases the probability of developing lung cancer.

Unlike linear regression, which is used to predict continuous variables, logistic regression is more suitable for binary outcomes in case-control studies.

2. Ability to Handle Multiple Factors:

Disease occurrence in case-control studies is influenced by various factors. Logistic regression can easily incorporate multiple independent variables, such as age, gender, and genetic factors, allowing for simultaneous analysis of their relationship with the disease and adjustment for confounders, accurately assessing the independent effects of each factor.

Compared to models that consider only single factors, logistic regression can comprehensively analyze the combined effects of multiple factors, aligning with real-world scenarios.

3. Flexible Data Distribution Requirements:

Logistic regression does not have strict requirements for data distribution and is applicable to various data types as long as the relationship between the independent variable and the dependent variable meets the logistic relationship assumption. This is advantageous in case-control study data that often do not meet normal distribution requirements.

4. Intuitive Interpretation of Results:

The regression coefficients in logistic regression have clear practical significance; their exponentiated form represents odds ratios (ORs), which intuitively reflect the change in the odds of disease occurrence for each unit change in the independent variable, making it easier for researchers and clinicians to understand and interpret the results.

Many other studies in case-control research have also opted for logistic regression as the statistical method. For instance, certain studies examining the relationship between smoking and lung cancer have utilized logistic regression for data analysis [1-5].

Comment 2:

“2. Have the findings been externally validated using an independent cohort or dataset?”

Response:

Thank you for your insightful question regarding the external validation of our findings. Unfortunately, we did not conduct an external validation using an independent cohort or dataset. We have acknowledged the limitations of our study and made the following updates to clarify these points in the limitation section: “For greater reliability and wider applicability, it is crucial that future research validates these results with independent cohorts.”.

Page 11; Line 224-225.

Comment 3:

“3. Despite adjusting for multiple confounders, the potential impact of unmeasured factors like dietary habits and physical activity cannot be ignored. These variables could significantly influence both fibrinogen levels and CRC risk. The authors should acknowledge this limitation and discuss its implications for their findings.”

Response:

Thanks for your great suggestion on improving the accessibility of our manuscript.

Thank you for your valuable feedback. We have revised the relevant section accordingly. The original sentence, "Thirdly, despite controlling for various factors, the potential for unmeasured variables could not be completely ruled out," has been changed to "Thirdly, despite controlling for various factors, unmeasured lifestyle factors (e.g., diet, physical activity), infections, and socioeconomic variables could still influence fibrinogen levels and CRC risk"

Page 10; Line 219-221.

Comment 4:

“4. The choice of statistical models (e.g., logistic regression) over others is not adequately justified. The rationale behind selecting specific methods and cut-off points for quartiles must be better explained, along with their clinical relevance.”

Response:

Thank you for your feedback. In case-control studies, logistic regression models are often preferred for several reasons:

1. Alignment with Research Objectives:

Case-control studies aim to explore the association between exposure and disease occurrence. Logistic regression can directly estimate the probability relationship between exposure and disease occurrence, obtaining odds ratios (ORs) that clearly demonstrate the impact of exposure factors on disease For instance, in studies examining the relationship between smoking and lung cancer, logistic regression can intuitively determine how much smoking increases the probability of developing lung cancer.

Unlike linear regression, which is used to predict continuous variables, logistic regression is more suitable for binary outcomes in case-control studies.

2. Ability to Handle Multiple Factors:

Disease occurrence in case-control studies is influenced by various factors. Logistic regression can easily incorporate multiple independent variables, such as age, gender, and genetic factors, allowing for simultaneous analysis of their relationship with the disease and adjustment for confounders, accurately assessing the independent effects of each factor.

Compared to models that consider only single factors, logistic regression can comprehensively analyze the combined effects of multiple factors, aligning with real-world scenarios.

3. Flexible Data Distribution Requirements:

Logistic regression does not have strict requirements for data distribution and is applicable to various data types as long as the relationship between the independent variable and the dependent variable meets the logistic relationship assumption. This is advantageous in case-control study data that often do not meet normal distribution requirements.

4. Intuitive Interpretation of Results:

The regression coefficients in logistic regression have clear practical significance; their exponentiated form represents odds ratios (ORs), which intuitively reflect the change in the odds of disease occurrence for each unit change in the independent variable, making it easier for researchers and clinicians to understand and interpret the results.

Many other studies in case-control research have also opted for logistic regression as the statistical method. For instance, certain studies examining the relationship between smoking and lung cancer have utilized logistic regression for data analysis [1-5].

Additionally, the primary purpose of quartile analysis was to enhance the comprehensibility of the data and to support the results. However, the core findings of the manuscript are based on the restricted cubic spline analysis and the identified threshold effect.

We also performed tertile and quintile analyses using multivariate logistic regression, and the results were similar to those obtained from quartile analysis. These additional analyses further support the robustness and reliability of our findings. The results are as follows:

Variable Total, n Event, n (%) Crude model Model Ia Model IIb

OR (95% CI) P-value OR (95% CI) P-value OR (95% CI) P-value

Tertiles

T1 1191 36 (3) 1(Ref) 1(Ref) 1(Ref)

T2 1202 106 (8.8) 3.1 (2.11~4.57) <0.001 2.83 (1.88~4.25) <0.001 2.93 (1.89~4.52) <0.001

T3 1202 326 (27.1) 11.94 (8.37~17.03) <0.001 8.51 (5.84~12.41) <0.001 6.53 (4.22~10.11) <0.001

P for trend 3595 468 (13) 3.59 (3.06~4.2) <0.001 2.95 (2.49~3.48) <0.001 <0.001

Quintiles

Q1 717 13 (1.8) 1(Ref) 1(Ref) 1(Ref)

Q2 710 36 (5.1) 2.89 (1.52~5.5) 0.001 2.58 (1.33~5.03) 0.005 2.53 (1.27~5.04) 0.008

Q3 705 59 (8.4) 4.95 (2.69~9.1) <0.001 4.24 (2.25~7.98) <0.001 4.33 (2.23~8.42) <0.001

Q4 741 112 (15.1) 9.64 (5.38~17.29) <0.001 7.29 (3.97~13.4) <0.001 7.51 (3.91~14.43) <0.001

Q5 722 248 (34.3) 28.33 (16.03~50.08) <0.001 19.22 (10.62~34.8) <0.001 13.75 (7.11~26.59) <0.001

P for trend 3595 468 (13) <0.001 <0.001 <0.001

OR, odds ratio; CI, confidence interval; Q, quartile; ALB, albumin; GGT, gamma-glutamyl transferase; ALP, alkaline phosphatase; AST, aspartate aminotransferase; ALT, alanine aminotransferase; ChE, cholinesterase; TBA, total bile acid; TBIL, total bilirubin; HDL, high density lipoprotein; LDL, low density lipoprotein; GLU, glucose; CREA, creatinine; TP, total protein; Lp(a), lipoprotein(a); PLT, platelets; WBC, white blood cell; PT-INR, international normalized ratio of prothrombin time; TT, thrombin time; APTT, activated partial thromboplastin time; CRC, colorectal cancer.

aAdjusted for age and sex. bAdjusted for Model I + weight, family history of CRC, drinking status, smoking status, ALB, ALT, AST, ALP, GGT, ChE, TBIL, TBA, GLU, CREA, urea, HDL, LDL, TP, Lp(a), PLT, WBC, neutrophil count, PT-INR, APTT, TT, hypertension, and liver disease.

Comment 5:

“5. What specific clinical applications or changes in practice could arise from these findings? How do the authors envision their results impacting patient care or public health strategies?”

Response:

Thank you for reviewing our work and for your valuable feedback. We have added this section to the discussion section of the manuscript:

“We discovered a non-linear, inverse L-shaped relationship between fibrinogen levels and CRC risk, pinpointing a critical threshold at 3.794 g/L. This finding provides significant evidence for targeting fibrinogen levels as a means of reducing CRC risk through clinical interventions. Notably, when fibrinogen levels are controlled below 3.794 g/L, CRC risk decreases substantially as fibrinogen levels drop. The threshold offers vital evidence for managing fibrinogen levels. Nevertheless, the potential diagnostic utility of fibrinogen can be enhanced when used in conjunction with other tumor markers such as carcinoembryonic antigen and prealbumin, possibly leading to improved diagnostic performance in CRC [6]. Future clinical trials and studies are needed to validate the effectiveness of fibrinogen as a diagnostic marker and to further explore its role in the adenoma-carcinoma sequence.”

Page 10; Line 206-215.

Reply to Reviewer #2

Thank you again for your positive comments and valuable suggestions to improve the quality of our manuscript. And we hope the revised manuscript could be acceptable for you.

Comments:

“This manuscript provides valuable insights into the association between fibrinogen and CRC in the Chinese population. However, several aspects require further elaboration to enhance the robustness and interpretability of the findings.

1. The manuscript sometimes implies a causal relationship between fibrinogen and CRC (e.g., "fibrinogen promotes cancer progression"). However, as this is a cross-sectional and retrospective study, causality cannot be inferred. Reframe the language throughout the manuscript to emphasize association rather than causality.

2. Although the study adjusts for numerous confounders (e.g., BMI, smoking, liver disease), certain lifestyle factors (diet, exercise), acute or chronic infective conditions, and socioeconomic variables are missing. These may influence both fibrinogen levels and CRC risk.

3. The study highlights a linear association between fibrinogen and CRC risk. However, non-linear relationships or threshold effects are not thoroughly investigated. Consider performing and reporting restricted cubic spline analyses to assess potential non-linear associations. This could reveal critical fibrinogen thresholds that significantly elevate CRC risk. Furthermore, if non-linear patterns are observed, discuss how clinical guidelines might use fibrinogen cut-off values to stratify CRC risk in patients.

4. Expand the discussion by integrating more references on the thrombin-fibrinogen axis or tumor microenvironment, and how fibrinogen promotes tumor growth, angiogenesis, and metastasis in CRC patients.

5. Abbreviations like ALP, AST, WBC, etc. used earlier in the text should be consistently applied in the subsequent sensitivity analysis section.

6. The link between fibrinogen and colorectal cancer might be attributed to chronic inflammation and the changes in the body's internal environment caused by tumors, such as coagulation issues. This is similar to other inflammatory markers like C-reactive protein, which also lack specificity. The studies mentioned by the author in the discussion section only show a connection between fibrinogen and colorectal cancer, but they don't establish that high levels of fibrinogen are a risk factor for the disease. Therefore, the author should provide a clear

---

## [Decision Letter · Decision Letter 1]

12 Feb 2025

PONE-D-24-36493R1Fibrinogen and colorectal cancer: A study of Chinese populationPLOS ONE

Dear Dr. Wang,

Thank you for submitting your manuscript to PLOS ONE. After careful consideration, we feel that it has merit but does not fully meet PLOS ONE’s publication criteria as it currently stands. Therefore, we invite you to submit a revised version of the manuscript that addresses the points raised during the review process.

We look forward to receiving your revised manuscript.

Kind regards,

José M. Alvarez-Suarez

Academic Editor

PLOS ONE

Journal Requirements:

Reviewers' comments:

Reviewer's Responses to Questions

**Comments to the Author**

1. If the authors have adequately addressed your comments raised in a previous round of review and you feel that this manuscript is now acceptable for publication, you may indicate that here to bypass the “Comments to the Author” section, enter your conflict of interest statement in the “Confidential to Editor” section, and submit your "Accept" recommendation.

Reviewer #1: All comments have been addressed

Reviewer #2: All comments have been addressed

2. Is the manuscript technically sound, and do the data support the conclusions?

Reviewer #1: Yes

Reviewer #2: Yes

3. Has the statistical analysis been performed appropriately and rigorously? 

Reviewer #1: No

Reviewer #2: Yes

4. Have the authors made all data underlying the findings in their manuscript fully available?

Reviewer #1: Yes

Reviewer #2: Yes

5. Is the manuscript presented in an intelligible fashion and written in standard English?

Reviewer #1: Yes

Reviewer #2: Yes

6. Review Comments to the Author

Reviewer #1: The revisions have strengthened the manuscript, particularly in clarifying methodological choices, acknowledging limitations, and expanding the discussion. Below are my remaining observations and suggestions for further improvement:

1. The authors provided a detailed rationale for using logistic regression, including alignment with study objectives, handling of confounders, and intuitive interpretation of ORs. Tertile analyses further support the robustness of quartile-based findings.

2. The study involves multiple subgroup analyses and sensitivity analyses, which increase the risk of Type I errors (false positives). The authors do not mention any adjustments for multiple testing.

3. The authors should provide more concrete recommendations for clinical practice, such as specific scenarios where fibrinogen monitoring could be beneficial.

4. The sensitivity analysis with complete cases (n=1,695) shows consistent results, but the manuscript should report the proportion of missing data and imputation methods in the Methods section.

5. Ensure the restricted cubic spline plot clearly labels the threshold (3.794 g/L) and includes confidence intervals.

Reviewer #2: The author has made sufficient revisions according to the reviewer's requirements, and I recommend it to be published

7. PLOS authors have the option to publish the peer review history of their article (what does this mean? ). If published, this will include your full peer review and any attached files.

**Do you want your identity to be public for this peer review?** For information about this choice, including consent withdrawal, please see our Privacy Policy .

Reviewer #1: No

Reviewer #2: **Yes: ** WEI WANG

---

## [Author Response · Author response to Decision Letter 1]

22 Feb 2025

Reply to Reviewer #1

Thank you very much for your time involved in reviewing the manuscript and your very encouraging comments on the merits.

Comments:

“The revisions have strengthened the manuscript, particularly in clarifying methodological choices, acknowledging limitations, and expanding the discussion. Below are my remaining observations and suggestions for further improvement:

1. The authors provided a detailed rationale for using logistic regression, including alignment with study objectives, handling of confounders, and intuitive interpretation of ORs. Tertile analyses further support the robustness of quartile-based findings.

2. The study involves multiple subgroup analyses and sensitivity analyses, which increase the risk of Type I errors (false positives). The authors do not mention any adjustments for multiple testing.

3. The authors should provide more concrete recommendations for clinical practice, such as specific scenarios where fibrinogen monitoring could be beneficial.

4. The sensitivity analysis with complete cases (n=1,695) shows consistent results, but the manuscript should report the proportion of missing data and imputation methods in the Methods section.

5. Ensure the restricted cubic spline plot clearly labels the threshold (3.794 g/L) and includes confidence intervals.”

Comment 1:

“1. The authors provided a detailed rationale for using logistic regression, including alignment with study objectives, handling of confounders, and intuitive interpretation of ORs. Tertile analyses further support the robustness of quartile-based findings.”

Response:

Thank you for your positive feedback. We appreciate your recognition of our work.

Comment 2:

“2. The study involves multiple subgroup analyses and sensitivity analyses, which increase the risk of Type I errors (false positives). The authors do not mention any adjustments for multiple testing.”

Response:

Thank you for your valuable feedback. We appreciate your suggestion regarding the need for multiple testing correction.

We would like to clarify that the interaction P-values for all subgroups (e.g., sex, age, HLP, hypertension, ischemic cerebrovascular disease, and CHD) were greater than 0.05, indicating no statistically significant differences in the relationship between fibrinogen and CRC across these subgroups. As such, we believe that the association is consistent across different subgroups, and there is no evidence of significant effect modification. In light of these findings, even if a Bonferroni correction were applied, the results would remain unchanged, as none of the interaction P-values would reach the adjusted significance threshold.

Comment 3:

“3. The authors should provide more concrete recommendations for clinical practice, such as specific scenarios where fibrinogen monitoring could be beneficial.”

Response:

Thanks for your great suggestion on improving the accessibility of our manuscript. We have supplemented the portion of the discussion that deals with this aspect. This paragraph is specified below:

We discovered a non-linear, inverse L-shaped relationship between fibrinogen levels and CRC risk, identifying a critical threshold at 3.794 g/L. This finding provides strong evidence supporting the use of fibrinogen levels in clinical interventions aimed at reducing CRC risk. Notably, when fibrinogen levels are controlled below 3.794 g/L, CRC risk decreases significantly as fibrinogen levels decline, highlighting the importance of this threshold in managing fibrinogen levels. Additionally, the diagnostic utility of fibrinogen may be enhanced when combined with other tumor markers, such as carcinoembryonic antigen and prealbumin, which could potentially improve CRC diagnostic performance [44]. Future clinical trials and studies are needed to further explore the role of fibrinogen in the adenoma-carcinoma sequence and validate its effectiveness as a diagnostic biomarker. Upon such validation, monitoring fibrinogen levels could become a valuable risk assessment tool prior to colonoscopy in CRC screening, owing to its efficiency, speed, and non-invasive nature. When elevated fibrinogen levels are detected, they should be assessed in conjunction with other risk factors to evaluate CRC risk, and colonoscopy should be employed for screening high-risk individuals.

Page 10; Line 208-220.

Comment 4:

“4. The sensitivity analysis with complete cases (n=1,695) shows consistent results, but the manuscript should report the proportion of missing data and imputation methods in the Methods section.”

Response:

Thank you for your valuable comments. Based on your suggestion, we have modified the following sentence in the Methods section to clarify the methodology used in the sensitivity analysis:

The sentence has been changed from:

“Furthermore, this study used multiple imputations (five replications) and sensitivity analysis with complete cases.”

To:

“Missing data were handled using multiple imputations by chained equations with five replications. In addition, a sensitivity analysis using complete cases was performed to assess the robustness of the results.”

Page 5; Line 93-95.

Comment 5:

“5. Ensure the restricted cubic spline plot clearly labels the threshold (3.794 g/L) and includes confidence intervals.”

Response:

Thank you for your valuable feedback. In response to your suggestion, we have updated the restricted cubic spline plot to clearly label the threshold, now set at 3.794 g/L, with the associated confidence intervals (95% CI: 3.747–3.841).

Additionally, we have made corresponding updates in both the Results and Abstract sections to reflect these changes.

---

## [Decision Letter · Decision Letter 2]

28 Mar 2025

Fibrinogen and colorectal cancer: A study of the Chinese population

PONE-D-24-36493R2

Dear Dr. Wang,

We’re pleased to inform you that your manuscript has been judged scientifically suitable for publication and will be formally accepted for publication once it meets all outstanding technical requirements.

Kind regards,

José M. Alvarez-Suarez

Academic Editor

PLOS ONE

Additional Editor Comments (optional):

Reviewers' comments:

Reviewer's Responses to Questions

**Comments to the Author**

1. If the authors have adequately addressed your comments raised in a previous round of review and you feel that this manuscript is now acceptable for publication, you may indicate that here to bypass the “Comments to the Author” section, enter your conflict of interest statement in the “Confidential to Editor” section, and submit your "Accept" recommendation.

Reviewer #1: All comments have been addressed

2. Is the manuscript technically sound, and do the data support the conclusions?

Reviewer #1: Yes

3. Has the statistical analysis been performed appropriately and rigorously? 

Reviewer #1: Yes

4. Have the authors made all data underlying the findings in their manuscript fully available?

Reviewer #1: Yes

5. Is the manuscript presented in an intelligible fashion and written in standard English?

Reviewer #1: Yes

6. Review Comments to the Author

Reviewer #1: The revisions have strengthened the manuscript, particularly in clarifying methodological choices, acknowledging limitations, and expanding the discussion. The manuscript has been well revised and could be accepted.

7. PLOS authors have the option to publish the peer review history of their article (what does this mean? ). If published, this will include your full peer review and any attached files.

**Do you want your identity to be public for this peer review?** For information about this choice, including consent withdrawal, please see our Privacy Policy .

Reviewer #1: No

---

## [Editor Report · Acceptance letter]

PONE-D-24-36493R2

PLOS ONE

Dear Dr. Wang,

I'm pleased to inform you that your manuscript has been deemed suitable for publication in PLOS ONE. Congratulations! Your manuscript is now being handed over to our production team.

Kind regards,

on behalf of

Professor José M. Alvarez-Suarez

Academic Editor

PLOS ONE